# Changes in Plasma Itaconate Elevation in Early Rheumatoid Arthritis Patients Elucidates Disease Activity Associated Macrophage Activation

**DOI:** 10.3390/metabo10060241

**Published:** 2020-06-10

**Authors:** Rónán Daly, Gavin Blackburn, Cameron Best, Carl S. Goodyear, Manikhandan Mudaliar, Karl Burgess, Anne Stirling, Duncan Porter, Iain B. McInnes, Michael P. Barrett, James Dale

**Affiliations:** 1Glasgow Polyomics, University of Glasgow, Glasgow G61 1BD, UK; Ronan.Daly@glasgow.ac.uk (R.D.); Gavin.Blackburn@glasgow.ac.uk (G.B.); Manikhandan.Mudaliar@glasgow.ac.uk (M.M.); Karl.Burgess@ed.ac.uk (K.B.); Michael.Barrett@glasgow.ac.uk (M.P.B.); 2Institute of Infection, Immunity and Inflammation, University of Glasgow, 120 University Place, Glasgow G12 8TA, UK; c.best.1@research.gla.ac.uk (C.B.); Carl.Goodyear@glasgow.ac.uk (C.S.G.); Duncan.Porter@glasgow.ac.uk (D.P.); Iain.McInnes@glasgow.ac.uk (I.B.M.); 3Institute of Biodiversity Animal Health and Comparative Medicine, University of Glasgow, Bearsden Road, Glasgow G61 1QH, UK; 4Institute of Quantitative Biology, Biochemistry and Biotechnology, The University of Edinburgh, Edinburgh EH9 3FF, UK; 5Department of Rheumatology, Gartnavel General Hospital, Glasgow G12 0YN, UK; astirling10@icloud.com; 6Department of Rheumatology, Wishaw General Hospital, 50 Netherton Street, Wishaw, North Lanarkshire ML2 0DP, UK

**Keywords:** rheumatoid arthritis, DMARD, macrophage, itaconate, inflammation, liquid chromatography–mass spectrometry (LC-MS), biomarker discovery, tricarboxylic acid (TCA) cycle, precision medicine

## Abstract

Changes in the plasma metabolic profile were characterised in newly diagnosed rheumatoid arthritis (RA) patients upon commencement of conventional disease-modifying anti-rheumatic drug (cDMARD) therapy. Plasma samples collected in an early RA randomised strategy study (NCT00920478) that compared clinical (DAS) disease activity assessment with musculoskeletal ultrasound assessment (MSUS) to drive treatment decisions were subjected to untargeted metabolomic analysis. Metabolic profiles were collected at pre- and three months post-commencement of nonbiologic cDMARD. Metabolites that changed in association with changes in the DAS44 score were identified at the three-month timepoint. A total of nine metabolites exhibited a clear correlation with a reduction in DAS44 score following cDMARD commencement, particularly itaconate, its derived anhydride and a derivative of itaconate CoA. Increasing itaconate correlated with improved DAS44 score and decreasing levels of C-reactive protein (CRP). cDMARD treatment effects invoke consistent changes in plasma detectable metabolites, that in turn implicate clinical disease activity with macrophages. Such changes inform RA pathogenesis and reveal for the first time a link between itaconate production and resolution of inflammatory disease in humans. Quantitative metabolic biomarker-based tests of clinical change in state are feasible and should be developed around the itaconate pathway.

## 1. Introduction

Rheumatoid arthritis (RA) is a chronic, destructive, immune-mediated inflammatory condition that predominantly affects synovial joints. In genetically susceptible individuals, mucosal exposure to external stimuli (e.g., cigarette smoke) triggers persistent systemic autoimmunity, and subsequent inflammatory cell articular recruitment, leading to tissue damage [1]. Constitutional features, such as weight loss, malaise or fever are prevalent in RA patients and RA patients exhibit an increased resting metabolic rate [2], which may in part be related to increased immune cell activation and turnover. Many data now suggest that circulating leukocyte subsets exhibit altered phenotypic and functional properties in the context of RA [3]. Chronic synovitis is associated with angiogenesis and, consequently, increased mediator release, e.g., prostanoids and chemokines that may be detected in the circulation. Cardiometabolic disease is a common comorbidity, reflected particularly in dysregulation of lipid metabolism [4] and has been attributed to an interaction between conventional risk factor pathways and systemic pro-inflammatory cytokines [1]. Thus, it is possible that changes in disease activity state may be reflected in measurable changes in biochemical activity that is demonstrated through detailed characterisation of metabolite profiles. 

Metabolomic technologies provide a detailed description of the relative abundance of individual metabolites within a single tissue or biological system [5,6]. At an individual level, these metabolomic “signatures” are the final expression of a complex process of gene-environmental interactions, gene and inflammatory cell activation and protein synthesis [5]. Analysing metabolomic profiles across a group of individuals can offer insights into disease pathogenesis when common associations with clinical phenotype emerge. The increased ability to measure quantities of a wide range of different metabolites has permitted detailed description of metabolic profiles across a range of complex, polygenic disorders [7]. Nuclear Magnetic Resonance (NMR) and Liquid Chromatography coupled to Mass Spectrometry (LC-MS) based metabolomics are being increasingly employed to understand the biochemical changes associated with RA [5,8,9,10,11,12,13,14,15]. NMR metabolomics is capable of directly providing identity and absolute concentration for a small number of metabolites. LC-MS is not capable of directly providing absolute concentrations, but can have a larger coverage of the metabolome. For example, plasma metabolic profiles, obtained using ^1^H-NMR, differentiated patients with different RA disease activity and showed treatment with TNF-α inhibitors modified the baseline metabolic profiles associated with active RA to resemble those of patients in remission [16]. Further, serum metabolite profiles obtained using ^1^H-NMR at baseline, and at 24 weeks after treatment, also distinguished responders from non-responders to methotrexate treatment [17]. 

Herein, we used an LC-MS platform to characterise changes in the plasma metabolomic profile in newly diagnosed RA patients commencing first-line nonbiologic conventional disease-modifying antirheumatic (cDMARD) therapy. Through an untargeted approach, we aimed to determine whether the levels of individual metabolites correlated to disease activity following initiation of treatment and whether changes in disease activity were also reflected in changes in the level of certain metabolites. 

## 2. Results

### 2.1. Study Population 

The baseline and 3-month characteristics of the patients included in this study are summarised in Table 1. In this study, DAS44 was calculated using erythrocyte sedimentation rate (ESR). At baseline, 75 patients commenced methotrexate and four commenced sulphasalazine. After three months of follow-up, we detected a significant improvement in disease activity, with a mean reduction in DAS44 from baseline of 2.1 (SD 1.4). Thirty-five patients were exposed to corticosteroid treatment (one oral, nine intra-articular only, 19 intramuscular only, six intra-articular and intramuscular) prior to donating research blood samples.

### 2.2. Metabolomic Analysis

Plasma from patients was subjected to untargeted metabolomics analysis using LC-MS [18]. A PCA plot (Appendix A) reveals no appreciable global separation from baseline to three months. Nevertheless, some individual peak changes were evident and those relating to the biggest differentiators in the PCA loadings were checked against lists of common contaminants [19] and assessed chromatographically as a safeguard against the observed separation being due to a sample handling/processing factor. 

### 2.3. Metabolite Comparisons

Comparisons were performed between individual peaks at baseline and three months, to see if there were any significant differences. These comparisons used a basic *t*-test to calculate a *p*-value and log fold-change difference. The *p*-values were used to control the false discovery rate, by calculating *q*-values. Those differences with a value of *q* < 0.05 were reported as significant. Out of 3042 peaks in the dataset, 464 were reported as different between baseline and three months. These values can be seen in the accompanying spreadsheet (Appendix A). 

### 2.4. Relationship between Changes in Metabolite Levels and Changes in DAS44 

To determine whether changes in disease activity were matched by changes in metabolomic profile, differences in DAS44 and metabolite levels between time points were calculated, for all patients. Linear regression was then performed, regressing DAS44 change on change in peak levels, for the baseline to three months. Each set of regressions admitted an effect size and *p*-value. These *p*-values were used to control the false discovery rate by calculating *q*-values. 

Between baseline and three months, nine significant effects were found for values of *q* < 0.05. A volcano plot of all the peaks is shown in Figure 1, with information on the significant peaks in Table 2. Indicative plots of these data and models are shown in Figure 2. For example, looking at the model for Peak.n.724, the slope of the line is −0.5, which indicates that a doubling/halving of the concentration after treatment is associated with an extra change in DAS44 downwards/upwards of approximately 0.5. This extra change is on top of the average DAS44 change in the whole population. Once identified as significant effects, these signals were manually assessed to determine peak quality and identity (where possible). 

Foremost among those metabolites associated with the decline in DAS44 score were itaconate (*m*/*z* = 130.0266), a predicted itaconate anhydride (*m*/*z* = 112.016), and a fragment predicted as originating from itaconyl-CoA (130.0267) (Figure 3). Among the other metabolites were cholesterol, several peptides and a range of fatty acids. 

In order to verify the feature-by-feature analysis, a partial least-squares (PLS) analysis was performed on the peak change of the full set of features against change in DAS44. This analysis indicated one component and zero orthogonal components. The Q²Y metric was given as 0.125, with a *p*-value of 0.01 after 1000 permutations of the samples. The top nine features from this analysis corresponded exactly to the nine features given by the feature-by-feature analysis. 

These results were then checked against those peaks that had a significant difference between baseline and three months, to find those peaks where there was both a significant difference between peak levels in the population, and also where there was a correlation between the change in DAS44 and the difference in peak levels. There were three peaks with this property, Peak.p.133 (annotated as ser-ser-ala or gly-ser-thr), Peak.p.209 (annotated as LPC (15:0)) and Peak.n.1157 (mass of 281.7499, retention of 204 s). n.1157 was not matched to any known metabolite, it presents as a doubly charged [M − 2H^−^]^2−^ peak with a predicted formula of C_30_H_61_N_9_O. 

### 2.5. Itaconate and CRP Level Have Similar Predictive Power for Response

Blood CRP levels are measures of the acute phase response that have rapid change properties that can map with response to treatments. Accordingly, it is included as an indirect surrogate of immune cell activation and has been included in a number of composite disease activity measures such as DASCRP-28 and SDAI. In this study, change in CRP levels correlated positively with a change in DAS44 score (*r* = 0.41, *p* = 1.2 × 10^−3^), diminishing as disease activity reduced. Conversely, change in itaconate correlated negatively with a change in DAS44 (*r* = −0.49, *p* = 9.6 × 10^−5^). Besides being correlated with DAS44, CRP and itaconate are also negatively correlated (*r* = −0.44, *p* = 4.9 × 10^−4^), as are ESR and itaconate (*r* = −0.38, *p* = 3.6 × 10^−3^). These associations are shown in Figure 4. As ESR is a factor in the calculation of DAS44, and ESR correlates with CRP, it is to be expected that there will be a correlation between CRP and DAS44. However, the higher significance of the relationship between Itaconate and DAS44 indicates that it might be more important than CRP in predicting DAS44 and hence patient status. It was also found that there was no apparent correlation between change in itaconate and change in the Health Assessment Questionnaire (HAQ) measure (*r* = −0.18, *p* = 1.7 × 10^−1^).

## 3. Discussion 

Metabolomics is emerging as a tool to identify biomarkers for disease, response to treatment and also indicators of pathogenesis that may offer routes for novel interventions. In the past few years, a number of studies using both NMR and mass-spectrometry-based approaches have been applied to study RA [5,6,7,8,9,10,11,12,13,14,15,16,17,21,22,23]. The use of an untargeted LC-MS platform has several benefits over the use of other platforms, such as NMR. Whilst the reproducibility and quantitation provided by NMR allows standardisation across laboratories, the technique is hampered by poor sensitivity and the inevitable overlap of strong signals (such as water) with weak ones (including many metabolites of interest). Mass spectrometry is capable of detecting, and identifying, a much broader range of metabolites and combining it with a suitable LC set up optimised for small, polar metabolites will provide a much more complete picture of the metabolome [18]. Mass spectrometry also allows for better identification of unknown metabolites; signals that do not match a known metabolite in any of the databases [24]. We, therefore, adopted this approach to seek novel biomarkers of state and response in patients with new-onset RA. 

Our results demonstrate a clear association between short-term changes in DAS44 scores and levels of a panel of nine metabolites. Principal amongst these is itaconate, its derived anhydride and a putative itaconate CoA derivative. Further, the correlation between the DAS44 score and itaconate is slightly more robust than that between the DAS44 score and CRP, indicating the itaconate might be as good a marker of improved patient status as CRP (both, however, remaining less good than DAS44 score alone). 

Itaconate has recently emerged as a primary moiety of interest in the pathogenesis of inflammatory disease and macrophage activation to an M1/inflammatory phenotype. Initially, an ability to inhibit succinate oxidation by succinate dehydrogenase was proposed as the key driver on its immunomodulatory effect [25]. Recently however, it has been shown that itaconate impacts directly on the anti-inflammatory transcription factor Nrf2 [26,27] to underpin its immunomodulatory activity. Itaconate is also the most pronounced marker of inflammatory arthritis in a murine model [28]. A recent study in the Tg197 murine model of inflammatory arthritis described itaconate to be a key marker of the disease, and elevated levels were found in afflicted mice that reversed upon TNF blockade with infliximab [28]. In other studies, itaconate appears to be involved in the regulation of inflammation, its elevation leading, ultimately, to suppression of the inflammatory response [29] and although functional roles in human inflammatory disease have yet to be reported, our studies indicate levels are elevated as inflammation is diminished. 

Our data reveals increasing itaconate associated with decreasing DAS score after cDMARD (primarily methotrexate) treatment, which is consistent with its anti-inflammatory role. However, the analysis in Tg197 mice indicated that elevated levels of itaconate were found during disease manifestation and these declined upon anti-inflammatory treatment with the anti-TNFα monoclonal antibody, infliximab. This observation appears to be contradictory to ours. However, a number of differences between the studies need to be considered. Our work used human plasma while hind limb tissue extracts and synovial fibroblast extracts were found to be the optimal material to find metabolite changes in the transgenic mouse model. Our first comparison point to baseline came at three months while the mouse comparator point was at six weeks and our study did not include a cohort of healthy controls to compare itaconate levels at the start of the experiment. Comparing cDMARD based therapy and biological-based therapy may be another confounding factor since different therapies work via different mechanisms to diminish inflammation and, therefore, may have differing influences on itaconate levels. Notwithstanding, having demonstrated a clear link between itaconate levels and response to treatment in RA, future studies using a wider range of variables will ultimately reconcile our understanding of the role of this pathway in inflammatory disease. 

This study cannot distinguish whether itaconate production is a result of improved condition due to treatment or is in itself responsible for that improved condition. Nor can it determine what impact exposure to specific treatments has on itaconate expression. In this study, the majority (95%) of patients received methotrexate as first-line DMARD; however, the results cannot distinguish whether the observed changes are related to the influence of changes in disease activity, or the direct effect of methotrexate, on itaconate expression. The initial observation requires validation in an independent RA cohort, in patients treated with other DMARD and biologic therapies, and within cohorts of patients with other inflammatory diseases. However, as the first indication that there is a clinically evident association between itaconate levels and disease activity levels, the need for additional work to understand if stimulating itaconate production pharmacologically offers a route to intervention in RA is of major importance. The findings suggest that further study of the itaconate pathway and macrophage activity may reveal additional important insights into immune function regulation and RA pathogenesis and may also reveal new, clinically relevant, markers of disease activity and treatment response. Finally, this study provides proof of concept that additional insights to disease pathogenesis can be identified through analysis techniques that combine highly detailed descriptions of metabolite expression with clinical data. 

## 4. Materials and Methods

### 4.1. Study Population

This study was conducted using clinical data and samples from 79 patients recruited to the Targeting Synovitis in Early Rheumatoid Arthritis (TaSER) study, a randomised clinical trial that compared the effectiveness of using either clinical (DAS28) or musculoskeletal ultrasound (MSUS) assessment of disease activity to drive an intensive early treatment strategy [30,31]. Briefly, at recruitment, all patients had active RA (DAS44 > 2.4) and both groups followed the same step-up sequence of DMARD escalation. In the DAS28 group, treatment was escalated until low disease activity was attained (DAS28 < 3.2) and in the MSUS group treatment was escalated until 1 or no joints of a limited 14 MSUS joint set exhibited any power Doppler (PD) signal. At the start of treatment patients were treated with methotrexate, or sulphasalazine if methotrexate was contraindicated, and combinations of intra-articular and intramuscular corticosteroids. Disease activity assessments, using the 44-joint disease activity score (DAS44), were conducted at baseline and every 3 months by a metrologist (AS) who was blinded to group allocation and treatment. The earliest that study group allocation could influence ongoing treatment was after 3 months of follow-up. The study protocol was approved by the West of Scotland Research Ethics Service (09/S0709/38) and was registered with ClinicalTrials.Gov (NCT00920478). All patients provided written consent to participate and for their disease activity results and tissue samples to be used for research purposes. All study activities were conducted in accordance with the Declaration of Helsinki. 

### 4.2. Sample Collection and Preparation

All patients donated additional blood samples for research purposes at baseline (Group A) and 3 months (Group B) using a standard operating procedure for sample harvest and processing. Blood was collected into lithium heparin vacutainers and stored on ice. Samples were centrifuged at 4 °C (1100× *g*, fixed angle rotor) within 4 hours of venipuncture and 500 uL aliquots of plasma were stored at −80 °C until required for analysis. 

### 4.3. Metabolomics

Samples were analysed by hydrophilic interaction liquid chromatography (HILIC)-mass spectrometry (LC-MS) (UltiMate 3000 RSLC (Thermo Fisher, San Jose, CA, USA) using a 150 × 4.6 mm ZIC-pHILIC column (Merck SeQuant, Umea, Sweden) running at 300 ll/min and Orbitrap Exactive (Thermo Fisher, San Jose, CA, USA) detection. Mass spectrometer parameters were: 50,000 resolving power in positive/negative switching mode. Electrospray ionisation (ESI) voltage was 4.5 kV in positive and 3 kV in negative modes. Buffers consisted of A: 20 mM ammonium carbonate in H_2_O and B: Merck SeQuant: acetonitrile. The gradient ran from 20% A: 80% B to 80% A: 20% B in 900 s, followed by a wash at 95% A: 5% B for 180 s, and equilibration at 20% A: 80% B for 300 s. 

The LC-MS data were processed using a combination of open-source tools run though R. Vendor-specific raw LC-MS files were converted into the mzXML open format using MSConvert from the proteowizard pipeline [32]. During conversion, the *m*/*z* data was centroided. Chromatographic peaks were extracted from the mzXML files using the centwave detection algorithm from XCMS and converted to PeakML files. Subsequently, PeakML files representing replicates were aligned and combined using mzMatch.R [33] after filtering out all peaks that were not reproducibly detected within groups. The combined PeakML files were subjected to additional intensity filtering, noise filtering and gap-filling to produce a set of reproducible peaks. These peaks were then corrected for instrument drift over time using an in-house Gaussian process regression algorithm modelled on the pooled samples. Peaks were manually checked for consistency and integrated using QuanBrowser (Thermo Fisher San Jose, CA, USA) where appropriate. Identifications were based on the Metabolomics Standards Initiative proposed minimum reporting standards. 

### 4.4. Statistical Analysis

Demographic and disease activity outcome data were collected from the TaSER study records. Tests of significant differences between peak levels were calculated using *t*-tests and controlled for by correcting the *p*-values for multiple testing by calculating *q*-values. Relationships between disease activity and metabolite levels were modelled using linear regressions and tests of significance were controlled by calculating *q*-values. Correlation coefficients were calculated using Pearson’s product-moment method. Full metabolome correlation analysis was performed using partial least-squares analysis of the full set of features on DAS44. Statistically highlighted features were manually assessed for peak shape to determine if they correspond to genuine metabolite related signals. Metabolite identification was carried out by first calculating an accurate molecular formula for the *m*/*z* value within 3 ppm. This formula was then compared to a list of authentic standards and assigned as a match if the retention time and peak shape were comparable. The list of authenticated standards is included as Appendix A. If not found in the authentic standards a putative assignment was made based on the retention time of the feature and the chemistry of the LC column using a curated list of 41,623 metabolites contained within the IDEOM database (http://mzmatch.sourceforge.net/ideom.html), as detailed by Creek et al. [34]. The study dataset has been uploaded to the online Metabolights repository (https://www.ebi.ac.uk/metabolights/MTBLS1583).

## 5. Conclusions

This study provides further support to the emerging view that itaconate has a regulatory role in inflammation and inflammatory disease. The study has shown that it is feasible to conduct analyses that combine sequential clinical outcome and metabolite expression data to identify novel biomarkers of disease state and gain additional insight into chronic, inflammatory disease pathogenesis. The results demonstrate an association between short-term, dynamic changes in rheumatoid arthritis disease activity and a panel of nine metabolites. Of these metabolites, changes in itaconate level particularly (and its derivatives) demonstrated a consistent negative correlation with changes in disease activity level (measured by DAS44-ESR) and CRP. These findings suggest a clinically relevant association between itaconate and rheumatoid arthritis disease activity and support further research into the role of itaconate in rheumatoid arthritis pathogenesis.

## Figures and Tables

**Figure 1 metabolites-10-00241-f001:**
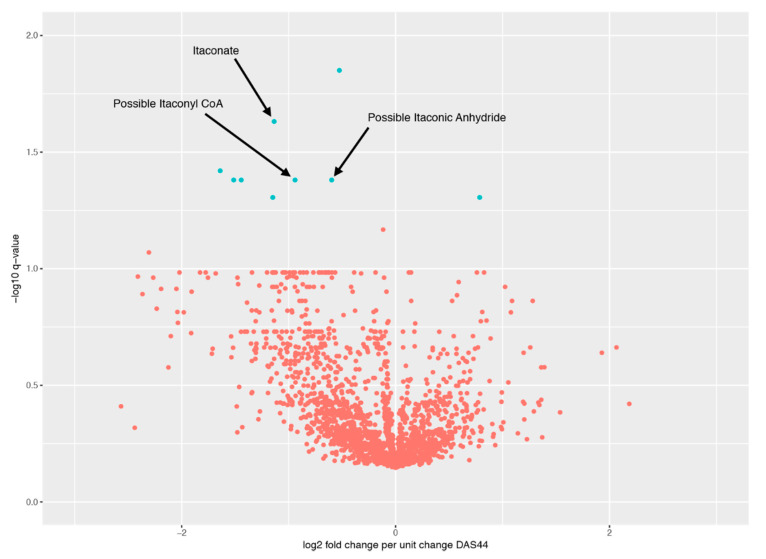
Volcano plot between baseline and three months. Blue points are significant peaks; orange points are not significant peaks.

**Figure 2 metabolites-10-00241-f002:**
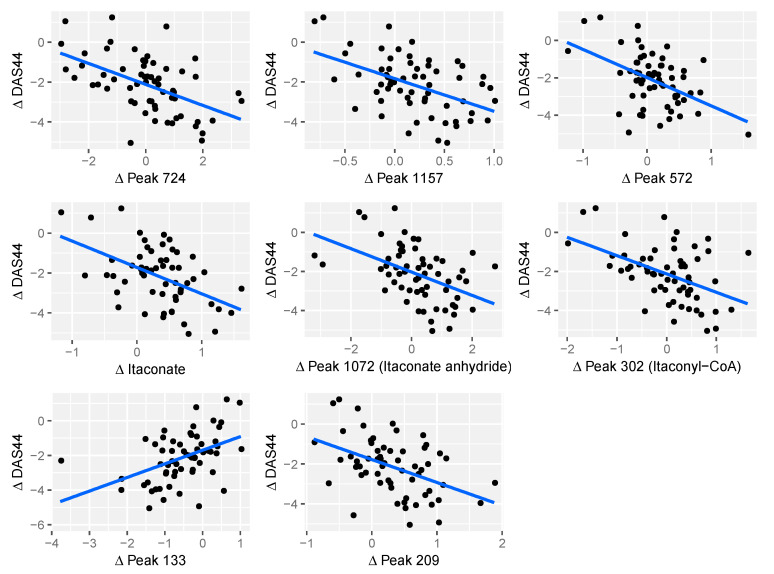
Scatter plots demonstrating change in DAS44 between baseline and 3 months vs. change in log2 peak intensity of 8 putative metabolites. The Itaconate peak (Peak 255) has been identified. Peaks 1072 and 302 have been given putative identities of Itaconate anhydride and Itaconyl-CoA, respectively.

**Figure 3 metabolites-10-00241-f003:**
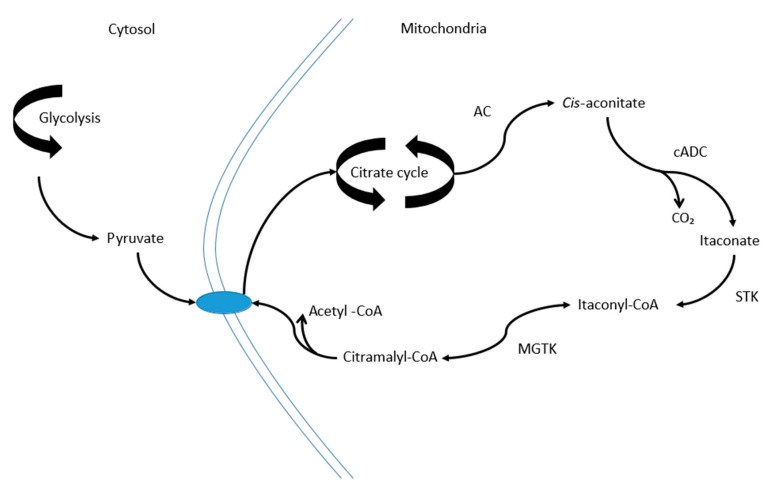
Metabolic pathway showing the production and itaconate via the TCA cycle and metabolism to pyruvate via itaconyl-CoA and citramalyl-CoA. *Cis*-aconitate is produced from citrate and isocitrate by aconitase (AC). This is converted to itaconate by *cis*-aconitate decarboxylase (cADC) which is then converted to itaconyl-CoA via succinate thiokinase (STK). Itaconyl-CoA is converted to citramalyl-CoA by methylglutaconyl-CoA hydratase (MGTK) which is then converted to pyruvate and acetyl CoA [20].

**Figure 4 metabolites-10-00241-f004:**
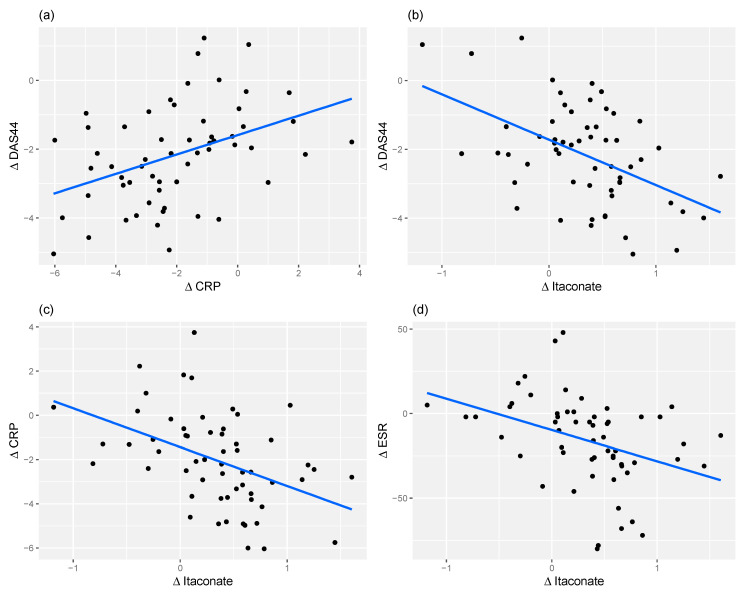
Scatter plots showing correlation between (**a**) change in CRP between baseline and 3 months vs. change in DAS44 between baseline and 3 months (*r* = 0.41); (**b**) change in itaconate between baseline and 3 months vs. change in DAS44 between baseline and 3 months (*r* = −0.49); (**c**) change in itaconate level between baseline and 3 months vs. change in CRP between baseline and 3 months (*r* = −0.44); (**d**) change in itaconate between baseline and 3 months vs. change in ESR between baseline and 3 months (*r* = −0.38).

**Table 1 metabolites-10-00241-t001:** Baseline characteristics and disease activity outcomes of metabolomics cohort (*n* = 79).

Demographics		
Female Sex, *n* (%)	54 (68%)	
Age (*y*)	56 ± 13	
Disease Duration (months)	5.3 ± 3.1	
Rheumatoid Factor Positive, *n* (%)	51 (65%)	
Anti-CCP Positive, *n* (%)	43 (53%)	
Plain X-ray Erosions, *n* (%)	26 (33%)	
**Disease Activity Outcomes**	**Baseline**	**3 Months**
DAS44	4.5 ± 1.2	2.3 ± 1.3
HAQ	1.5 ± 0.8	0.9 ± 0.9
ESR	36 ± 26	21 ± 21
CRP	42 ± 55	14 ± 25

Unless stated, values are mean ± SD. anti-cyclic citrullinated peptides (anti-CCP); disease activity score in 44 joints (DAS44); health assessment questionnaire (HAQ); erythrocyte sedimentation rate (ESR); C-reactive protein (CRP).

**Table 2 metabolites-10-00241-t002:** Annotated LC-MS peaks that have been differentially expressed across changing DAS44 scores.

Peak ID	Peak Change *q*-Value	*m*/*z*	RT (s)	Comments
Peak.n.724	0.0141	134.0579	477	Peak check passed. No ID
Peak.n.1157	0.0364	281.7499	204	Peak check passed. No ID
Peak.n.572	0.0364	466.3118	208	Peak check passed. Putative ID: cholesterol sulphate
Peak.n.302	0.0364	130.0267	435	Peak check passed. Putative ID: Itaconyl-CoA fragment, based on not matching standards for itaconic acid or isomers
Peak.n.1072	0.0364	112.0161	429	Peak check passed. Putative ID: ITACONIC-ANHYDRIDE
Peak.n.255	0.0234	130.0266	658	Peak check passed. Multiple peaks. Putative ID: Itaconate, Metabolomics Standards Initiative level 1, based on retention time and monoisotopic mass
Peak.n.1082	0.0364	467.3151	208	Peak check passed. Isotope of 572
Peak.p.133	0.0444	263.1115	616	Peak check passed. PEPTIDE_856, PEPTIDE_1100
Peak.p.209	0.004	481.3169	279	Peak check passed. Putative ID: lysoPC(15:0)

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
