# Peer review of "Changes in Plasma Itaconate Elevation in Early Rheumatoid Arthritis Patients Elucidates Disease Activity Associated Macrophage Activation"

_metabolites, 2020, doi:10.3390/metabo10060241_

Round 1

Reviewer 1 Report

Authors described that metabolomic analysis revealed that plasma Itaconate was associated with treatment response of RA. The point of focus is interesting however, several numbers of concerns need to be addressed by authors.

#1. Almost all patient were treated with MTX. The possibility that MTX affect itaconate metabolism should be discussed.

#2 As mentioned in discussion section by authors, in this study healthy control or disease control group (other inflammatory disease) are needed. Otherwise, it is unclear whether itaconate metabolism is related with rheumatoid arthritis or not.

#3 The meaningful of Figure 4 is unclear. The significant relation of DAS44 and CRP is obvious in RA. So, the strong relation of CRP with itaconate is naturally when DAS44 is related to itaconate. The authors should describe cleary why this analysis was performed.

Author Response

We thank you for your thoughtful and positive comments. They have allowed us to revise the manuscript in a positive manner, add extra useful information and clear up any misunderstandings that could have befallen the reader

We hope these changes have addressed your concerns.

Please see the attachment for the full response.

Reviewer 2 Report

In this study, the authors have compared the plasma metabolite profile in newly diagnosed rheumatoid arthritis patients before and at 3 months following the start of treatment with disease-modifying anti-rheumatoid arthritis drugs. They report that plasma levels of itaconate and its derivatives correlated negatively with DAS44 and CRP. They suggest that itaconate is clinically relevant marker RA disease activity and treatment response.

General comment:  The data are sound and well presented.

Minor comments:

The title is too general since 95% of the newly diagnosed rheumatoid arthritis received methotrexate. The conclusion applies to the methotrexate treatment and not for medications with other conventional disease-modifying anti-rheumatoid arthritis drugs. I would suggest “…following successful methotrexate treatment…”

Paragraph 4.2: No data at 18 months follow-up are shown.  This is not essential information.

Author Response

We thank you for your thoughtful and positive comments. They have allowed us to revise the manuscript in a positive manner, add extra useful information and clear up any misunderstandings that could have befallen the reader.

We hope these changes have addressed your concerns.

Please see the attachment for the full response.

Reviewer 3 Report

This manuscript by Daly et al. studies the change in metabolite profile after 3-month treatment with DMARDS in a cohort of subjects with early RA. The authors conclude that itaconate and its derivatives associate with an improvement in DAS score after treatment, shading new light on itaconate as an anti-inflammatory factor in RA.

The manuscript is interesting, novel and well written. I have some suggestions to improve it:

- is the DAS44 score you used in the analyses based on CRP or ESR? If it is based on CRP (please state it in Methods), then Fig. 4a is pleonastic as it shows a positive association between CRP and a score including CRP itself. However, I see your point in showing this figure, as you state in the discussion that “the correlation between DAS44 score and itaconate is slightly more robust than that between DAS44 score and CRP, indicating the itaconate might be as good a marker of improved patient status as CRP”. I would suggest you expanding this part in Results explaining why you looked at the association between CRP and DAS44.

- Fig.4b and c are important as they point out that an increase in itaconate associates with lower RA disease activity. Is there any association between itaconate and ESR? If there is, this can strengthen the hypothesis than higher itaconate levels associate with lower inflammation in RA. Is there any association between itaconate and HAQ?

- could you please show the follow-up characteristics of the cohort, i.e. DAS44, HAQ, ESR and CRP 3 months after treatment?

- please describe in a few words what NMR and LC-Ms are in the introduction.

Author Response

We thank you for your thoughtful and positive comments. They have allowed us to revise the manuscript in a positive manner, add extra useful information, and clear up any misunderstandings that could have befallen the reader.

We hope these changes have addressed your concerns.

Please see the attachment for the full response.

Round 2

Reviewer 1 Report

I am curious for itaconate metabolism in control group.  The additional experiment are needed by authors in near future. However, regarding other question, the author replied adequately.

Author Response

Point 1:

I am curious for itaconate metabolism in control group.  The additional experiment are needed by authors in near future. However, regarding other question, the author replied adequately.

Response 1:

We acknowledge that it would be useful for future work involving itaconate metabolism to involve a control group to investigate differences with the study group.

We thank you for your positive remarks and helpful comments which contributed to the revised manuscript being improved. 

Reviewer 3 Report

I am happy with the changes in the manuscript. I do not have further comments

Author Response

We thank you for your positive comments and helpful feedback which helped improve the manuscript following its revision.